# Electrochemically Reduced Graphene Oxide Covalently Bound Sensor for Paracetamol Voltammetric Determination

**DOI:** 10.3390/ijms26094267

**Published:** 2025-04-30

**Authors:** Amaya Paz de la vega, Fabiana Liendo, Bryan Pichún, Johisner Penagos, Rodrigo Segura, María Jesús Aguirre

**Affiliations:** 1Department of Chemistry of Materials, Faculty of Chemistry and Biology, Universidad de Santiago de Chile (USACH), Santiago 9170022, Chile; amaya.pazdelavega@usach.cl (A.P.d.l.v.);; 2Millennium Institute on Green Ammonia as Energy Vector—MIGA (ICN2021_023), Santiago 7820436, Chile

**Keywords:** electrochemical grafting, 4-nitroaniline, reduced graphene oxide, paracetamol, square wave voltammetry, pharmaceutical tablets

## Abstract

Designing a highly sensitive and efficient functionalized electrode for precise drug analysis remains a significant challenge. In this work, an electrochemical sensor based on a glassy carbon electrode (GCE) modified with phenyl diazonium salts (ph) and electrochemically reduced graphene oxide (ERGO), labeled GCE/ph/ERGO, was developed for the detection of paracetamol (PAR) in pharmaceutical matrices using square wave voltammetry (SWV). The modified electrode was characterized by scanning electron microscopy (SEM), electrochemical impedance spectroscopy (EIS), and cyclic voltammetry (CV). Compared to the bare GCE, the GCE/ph/ERGO sensor demonstrated significantly improved conductivity and anodic current peak for PAR over two orders of magnitude higher, indicating a substantial enhancement in electrochemical performance. Under optimized conditions, the developed sensor exhibited a low detection limit of 18.2 nM and a quantification limit of 60.6 nM. Precision studies yielded relative standard deviations (RSDs) below 8%. The sensor demonstrated excellent selectivity in the presence of common pharmaceutical excipients and high accuracy in the analysis of generic pharmaceutical formulations, with results comparable to those obtained by the HPLC technique. These findings confirm the sensor’s reliability, stability, robustness, and suitability for routine analysis of PAR in pharmaceutical samples.

## 1. Introduction

Paracetamol (acetaminophen) is an over-the-counter medication that has an analgesic and antipyretic effect, used for the treatment of fever and mild to moderate pain [1]. It is well tolerated and has a minimal risk of gastrointestinal side effects. When misused or overdosed, paracetamol is also capable of inducing serious hepatic damage, which can lead to life-threatening liver necrosis [2]. Chronic use by itself is typically harmless, but when combined with other acetaminophen-containing medications, it presents a substantial risk of toxicity [3]. In fact, acetaminophen toxicity is the second leading cause of liver transplants worldwide [4,5].

These risks highlight the need for an effective quality control system in the pharmaceutical industry to ensure drug safety [6,7,8]. These quality control procedures are conducted to guarantee the identity, purity, and concentration of a particular pharmaceutical product, generating the desired pharmacological effect without causing adverse effects or overdose, to improve the safety and efficacy of these products, and to help keep the public healthy and safe.

The analysis of pharmaceutical compounds in aqueous matrices or pharmaceutical preparations is predominantly performed using chromatographic techniques, such as liquid chromatography (HPLC) coupled with mass spectrometers (MS) [9]. Although these techniques offer low detection limits and high sensitivity, their high acquisition and maintenance costs, along with the impossibility of using this technique “in situ”, have promoted the development of faster and more accessible techniques. Voltammetric techniques provide reliable results in terms of reproducibility and repeatability, low detection limits, easy implementation, and low-cost, low-maintenance instrumentation. Moreover, they are well-suited for use in portable systems [10,11,12,13,14]. However, most solid electrodes used for electroanalysis of organic molecules (e.g., glassy carbon, screen-printed, boron-doped diamond, and gold electrodes) have limitations, including surface fouling, small surface area, signal drift or sensitivity loss, and limited selectivity [15]. To overcome some of these disadvantages, chemically modified electrodes (CMEs) are being explored for sensitivity, selectivity, and stability improvements, which are based on the generation of a film of a modifying agent on the surface of an electrode, improving the sensitivity and selectivity of the measurement [16,17,18,19]. Various modifying agents have been used, ranging from metallic deposits to organic materials. They can be prepared by different methods, one of which is physisorption or “drop-coating”, which consists of applying a thin coating of a chemical compound, such as a carbon nanomaterial, by depositing consecutive drops of a suspension on the electrode surface and allowing the solvent to evaporate. Another surface modification technique is “electrografting”, an electrochemical reaction where a molecule, a diazonium salt, for example, is reduced or oxidized on the working electrode [20,21,22,23], granting great sensor stability resulting from the covalent bond between the modifier and the electrode surface. Electrochemically reduced graphene oxide (ERGO), a two-dimensional carbon nanomaterial that allows for easy access and synthesis, low cost, and simple functionalization [24,25,26], is among the nanomaterials used to modify an electrode. ERGO has been the subject of investigation in many technological fields over the past decades, including chemical sensors, biosensors/immunosensors, and nanocomposites. Electrochemical surfaces modified with ERGO have been extensively studied due to ERGO’s high surface area, excellent biocompatibility, and outstanding electrocatalytic and antifouling properties, making it effective for the detection of various molecules and biomolecules [19,27,28]. ERGO can be integrated onto electrode surfaces via several strategies, including physical adsorption or covalent attachment of functional groups [29]. Among these, diazonium chemistry offers a particularly robust approach for forming stable graphene-based films. Diazonium intermediates exhibit very low reduction potentials, enabling stable functionalization across a wide range of substrates without risk of oxidation. They are also water-compatible and can be generated from various aromatic amines [22,30,31]. These diazonium-derived organic layers exhibit excellent electron transfer properties, facilitating efficient charge transport between the electrode and the analyte. Reactions involving diazonium salts have been widely used to covalently functionalize metals, semiconductors, and carbon-based materials such as carbon nanotubes and graphene nanosheets [32].

Anchoring ERGO onto electrode surfaces via diazonium salt chemistry further enhances electrochemical sensor performance. ERGO improves electron transfer and conductivity, while its porous structure offers more active sites, thereby increasing sensitivity and current response. The strong covalent bonding provided by diazonium chemistry contributes to long-term stability and allows for surface tailoring to improve selectivity or antifouling behavior. Moreover, by carefully controlling the concentration of diazonium salts and the grafting conditions, multilayer formation can be avoided, preserving ERGO’s inherent conductivity and electroactivity. This composite strategy leads to reproducible, stable, and sensitive electrochemical platforms suitable for detecting organic analytes [33,34,35].

While many studies have focused on the covalent attachment of carboxyl-functionalized graphene using carbodiimide coupling chemistry [36,37], the use of diazonium chemistry to covalently anchor ERGO remains comparatively unexplored. Furthermore, although paracetamol detection has been widely studied using both carbon and graphene-based electrodes, the synergistic integration of phenyl diazonium salts with ERGO to construct high-performance sensing interfaces has received limited attention.

In this work, a glassy carbon electrode modified with electrochemically reduced graphene oxide via phenyl diazonium salt chemistry (GCE/ph/ERGO) was developed. The combination of ERGO and diazonium modification enhances sensor surface stability, conductivity, and sensitivity toward phenolic compounds such as paracetamol in pharmaceutical formulations, outperforming electrodes modified with physiadsorbed ERGO alone.

## 2. Results and Discussion

To obtain a highly conductive and sensitive sensing interface, we aimed to enhance the properties of ERGO through a controlled modification strategy. By introducing a conductive molecular anchor, we promoted ERGO attachment, avoiding agglomeration and preserving its electrochemical activity, improving the sensor’s performance in PAR detection.

### 2.1. Electrografting Process and Surface Fabrication

To ensure the formation of a controlled and thin aryl layer on the GCE surface, the electrografting conditions were carefully optimized: (i) a low concentration (1 mM) of the aromatic amine was used to limit aryl radicals generation and minimize multilayer formation; (ii) electrochemical grafting was performed via cyclic voltammetry using only two potential cycles within a narrow window, sufficient to initiate diazonium reduction without excessive radical accumulation or over-grafting; (iii) 4-nitroaniline was selected as the aromatic amine precursor due to the electron-withdrawing nitro group, which introduces electronic repulsion and steric hindrance, thereby reducing the likelihood of secondary aryl–aryl radical coupling and favoring monolayer formation [38,39].

In the first stage, p-nitrophenyl diazonium cations are generated in acid media using pNA and NaNO_2_ as precursors in the electrochemical cell. In the second stage, the covalent modification is carried out using electrografting or electrochemical reduction of the GCE surface with the p-nitrophenyl diazonium cations by CV. The molecule is reduced to a radical through the release of molecular nitrogen, and this aryl radical reacts with the nearest nucleophile or material, in this case, the electrode surface. In the third stage, the nitro groups on the surface are electrochemically reduced to amino groups to covalently anchor the nanomaterials to the sensor (see Figure 1A).

Figure 1B(i) shows the voltammogram obtained for the electrochemical reduction of p-nitrophenyl diazonium cations. The phenyl diazonium salt was generated “in situ” in the electrochemical cell. An irreversible reduction peak is observed around 0.01 V in the first cycle, consistent with the evidence reported regarding the formation of the diazonium radical using pNA by electrochemical methods (Equation (1)) [29,40]. This broad reduction peak is not observed in the second cycle, which indicates that the grafting process of the GCE surface with the p-nitrophenyl diazonium cations takes place in the first voltammetric cycle, and this grafted layer inhibits further reduction of nitrophenyl diazonium salts onto the GCE surface [41]. A redox peak at 0.33 V (oxidation) (Equation (3)) and 0.27 V (reduction) (Equation (2)) is observed in subsequent cycles and can be associated with the oxidation of hydroxyamino groups (Equation (3)) generated by a partial reduction of nitro groups (Equation (2)) present in the solution or the modified layer.(1)$$Ar-{NO}_{2}+e^{-}\to Ar-{NO}_{2}\;\cdot$$(2)$$Ar-{NO}_{2}+{4H}^{+}+{4e}^{-}\to Ar-NHOH+H_{2}O$$(3)$$Ar-NHOH↔Ar-NO+{2H}^{+}+{2e}^{-}$$(4)$$Ar-NHOH+{2H}^{+}+{2e}^{-}\to Ar-{NH}_{2}+H_{2}O$$

The modified p-nitrophenyl groups on the surface of GCE/pNO_2_ are electrochemically reduced to p-aminophenyl groups by CV in a protic solution (0.10 M KCl in 90:10 *v*/*v* H_2_O/EtOH). The voltammogram obtained (Figure 1B(ii)) shows a cathodic peak at −0.84 V, associated with the irreversible electroreduction of p-nitrophenyl to p-aminophenyl in a six-proton, six-electron process (through a mechanism 4 + 2e^−^) carried out in the modified layer (Equations (2) and (4)) [42,43].

In this work, we incorporated GO nanomaterials into the modified electrode through the diazotation of the amino groups present in the GCE/p-NH_2_ surface and the immediate anchoring of GO through drop-casting of a GO and NaNO_2_ acid solution, followed by electrochemical reduction of this GO layer.

Our investigation team previously reported [44] the optimal conditions for the irreversible electrochemical reduction of graphene oxide in aqueous solution. The process is conducted using cyclic voltammetry, applying five reduction cycles between 0.00 and −1.60 V. In Figure 1B(iii), a cathodic peak can be seen at −1.46 V, associated with the presence of epoxide, hydroxyl, and carboxylic acid groups in the GO structure. After the irreversible electroreduction of these oxygenated groups, we obtain electrochemically reduced graphene oxide on the electrode surface (GCE/ph/ERGO).

### 2.2. Morphological Characterization of the Modified Surface

The surface morphologies were investigated by SEM. Figure 2A shows SEM images of bare GCE. The surface morphology of GCE/p-NH_2_ (Figure 2B) shows no significant difference from bare GCE, indicating that the organic film formed does not influence the electrode surface morphology. In comparison, GCE/ph/ERGO (Figure 2C,D) shows modification of the nanomaterial in the form of white “wrinkled” layers on the surface.

### 2.3. Electrochemical Characterization of Modified Surface

All modified surfaces were characterized using CV and electrochemical impedance spectroscopy (EIS) in a solution containing the redox species ferro and ferricyanide (Fe(CN)_6_^3−/4−^). Figure 3A shows pronounced anodic and cathodic peaks for the GCE electrode characteristic of single-electron transfer of the redox couple (ΔE 0.07 V). The diazonium grafted surface (GCE/p-NO_2_) reported an increase in charge transfer resistance and suppression of the Fe(CN)_6_^3−/4−^ anodic and cathodic peaks, which confirmed successful surface modification consistent with a thin, covalently attached aryl layer. However, after electroreduction of the p-nitrophenyl groups to p-aminophenyl (GCE/p-NH_2_), the cathodic and anodic signals of the redox couple are recovered to some extent, with a decrease in current intensity compared to GCE (ΔE 0.10 V), which would indicate the generation of an organic film on the electrode surface that would impede electron transfer to a certain extent [45]. Incorporating GO by drop-casting and then electrochemically reducing it to ERGO (GCE/ph/ERGO) recovers the electronic transfer from the electrode surface, with an increase in the current intensity (ΔE 0.06 V) at levels comparable to the electrochemical response of an electrode modified with only ERGO (GCE/ERGO) (ΔE 0.08 V). This behavior is reflected in the results obtained using the EIS technique.

Figure 3B shows the Nyquist diagrams corresponding to GCE and the modified surfaces obtained from the impedance spectrum processed with a Randles equivalent electrical circuit. This model consists of the ionic resistance of the solution (Rs) connected in series to a constant phase element (CPE) that represents the capacitance of the double layer, which in turn is connected in parallel to a Warburg element (W) representing the diffusional resistance and resistance to electron transfer (Rct). The Rct is represented in the Nyquist diagram as the diameter of the semicircle formed and controls the rate of electron transfer at the interface of the redox probe between the solution and the electrode. Once the EIS results have been processed with the equivalent circuit, the Rct values can be obtained. The Rct value for the GCE electrode (Rct 211.27 Ω) increases substantially (Rct 3894 Ω) after modifying the surface with pNA in GCE/p-NO_2_. This response may result from the low conductivity of the p-nitrophenyl organic layer, which would hinder the electronic transfer of the redox couple to the electrode surface. After the electro-reduction of the p-nitrophenyl groups to p-aminophenyl, the charge transfer between the couple and the electrode surface becomes more efficient (Rct 338.57 Ω), probably due to the good conductivity of the p-aminophenyl layer. These results coincide with CV studies and are like those found in the bibliography [46,47,48,49]. A significant decrease in the Rct (Rct 3.24 Ω) value can be observed after attaching ERGO to the GCE/ph/ERGO modified surface due to the excellent electronic transfer of the nanomaterial, attributed to the partial restoration of the sp^2^ carbon network following the electroreduction [50].

### 2.4. Surface Area of Modified Electrodes

The surface coverage of diazonium salt-modified electrodes GCE/p-NO_2_ and GCE/p-NH_2_ was estimated using Equation (5), where Γ (mol cm^−2^) represents the surface coverage, A is the electrode geometric surface area (cm^2^), n is the number of electrons involved in the reaction, F is the Faraday constant (C mol^−1^), and Q is the total amount of charge (C) obtained by integrating the reduction peak recorded using CV. Assuming a four-electron process for the grafting of p-nitrophenyl diazonium cations (Equation (2)) and a six-electron process for the total reduction of p-nitrophenyl to p-aminophenyl groups (Equations (2) and (4)) [41], the surface coverage values calculated from the voltammetric charges (Figure 1A,B) were 1.41 × 10^−10^ mol cm^−2^ for GCE/p-NO_2_ and 1.74 × 10^−10^ mol cm^−2^ for GCE/p-NH_2_. These values are consistent with previously reported studies for surfaces modified with p-nitrophenyl groups and fall within the expected range for a close-packed monolayer of grafted phenyl groups [31,51,52,53,54].(5)$$\Gamma =\frac{Q}{nFA}$$

The electroactive surface area (ECSA) of GCE, GCE/ph/ERGO, and GCE/ERGO electrodes was estimated using the Randles–Sevcik equation (Equation (6)), where D is the diffusion coefficient of the redox probe (7.6 × 10^−6^ cm^2^ s^−1^), C the concentration of the redox species (5.0 µmol mL^−1^), n is the number of electrons transferred (n = 1), ν is the voltammetric scan rate (mV s^−1^), and A is the electroactive surface area (cm^2^) [44,55]. For a diffusion-controlled process, the peak current is linearly dependent on the square root of the scan rate (Appendix A). Based on the slopes obtained from this linear relation, the ECSA was calculated to be 0.10 cm^2^ for GCE, 0.13 cm^2^ for GCE/ph/ERGO, and 0.14 cm^2^ for GCE/ERGO. These results indicate a 30% increase in ECSA for GCE/ph/ERGO and a 40% increase for GCE/ERGO compared to bare GCE, confirming the successful surface modification and enhanced electrochemical activity of the modified electrodes.(6)$$Ip=2.69\times 10^{5}\;nACD^{1/2}v^{1/2}$$

### 2.5. Electrochemical Behavior of PAR at the Modified Electrode

The electrochemical response of the PAR analyte at the GCE and modified surfaces was studied in acetate buffer using square wave voltammetry (Figure 3C). From the voltammogram obtained, the oxidation peak of PAR can be observed at 0.47 V in concordance with the literature [56,57,58]. According to several studies, PAR exhibits quasi-reversible redox behavior, characterized by a well-defined anodic peak corresponding to the oxidation of the phenolic group to N-acetyl-p-benzoquinone imine (NAPQI) via a two-electron, two-proton process (Figure 1) [16,59,60,61].

Compared with GCE, GCE/p-NH_2_, and GCE/ERGO, the GCE/ph/ERGO modified electrode has the highest current response to PAR, which can be attributed to the increased electroactive surface area, high electron mobility, and favorable π–π interactions between paracetamol and the graphene sheets present in the modification. Based on the CV, EIS, and SWV results, the GCE/ph/ERGO electrode showed greater electronic transfer and current response to the analyte compared with non-modified GCE, diazonium-modified surfaces, and the GCE/ERGO electrode.

### 2.6. Effect of Experimental Variables

#### 2.6.1. Optimization of the Electrografting Process

To optimize the modifier concentration used in the electrografting process to find the best combination to preserve electronic transfer and generate an organic layer, a study of the effect of NaNO_2_ concentration on the diazotization process was performed by keeping the pNA concentration at 1 mM and varying the NaNO_2_ concentration between 1 and 3 mM. Appendix A shows the electrochemical characterizations of these GCE/p-NH_2_ electrodes using CV and EIS in a solution containing the redox species Fe(CN)_6_^3−/4−^. GCE and GCE/p-NH_2_ with 1 mM pNA and 1 mM NaNO_2_ (1:1) exhibit the behavior described for the redox couple (Figure 3A). However, as the NaNO_2_ concentration in the cell increases, it can be observed that the anodic and cathodic peaks of Fe(CN)_6_^3−/4−^ progressively decrease their current intensity and move to higher potentials (Appendix A), increasing the separation between peaks compared to GCE, which is consistent with the modification of the electrode and the generation of an organic layer on the electrode surface that acts as a barrier and prevents the electronic transfer to the electrode surface. The same behavior is observed in the Nyquist plots obtained using EIS (Appendix A) for the modified surfaces; as the concentration of sodium nitrite increases, the Rct value increases due to the generation of an organic multilayer on the surface. Using a 1:1 concentration in the electrografting process, the best electron transfer results for the modified surface are obtained.

#### 2.6.2. Optimization of Modified GO/NaNO_2_/HCl Agent

The effect of the volume of the GO modifier aliquot deposited on the surface of the modified electrode was evaluated on the current signal of 2 μM of PAR. For this purpose, the surface was modified with different aliquots of the 0.62 g L^−1^ GO solution, and its effect on the analyte current was studied. As shown in Appendix A, as the volume of the GO modifier aliquot increases, the current signal for PAR increases up to a maximum of 5 μL. At higher modifier volumes, a decrease in the PAR current signal is observed due to a saturation of the electrode surface, which would hinder the charge transfer at the electrode–solution interface.

#### 2.6.3. Optimization of Supporting Electrolyte and pH

As the electrochemical oxidation of PAR is a proton-coupled electron transfer reaction (see Figure 1), one would expect to observe a dependence on the supporting electrolyte and its pH.

The effect of different supporting electrolytes on the current signal of 2 μM PAR was evaluated using the following buffer systems: acetic acid/acetate (HAc/Ac^−^, pH 5), phosphate (PBS, pH 7), Britton–Robinson (pH 6.3), ammonia (pH 8.5), and 0.1 M sulfuric acid (H_2_SO_4_). According to the literature, PAR exhibits good electrochemical response in HAc/Ac^−^, PBS, and Britton–Robinson buffers at pH values below 8 [60,62,63,64], as it is more stable in slightly acidic to neutral conditions. These pH environments help prevent PAR degradation and improve electrode surface stability. For this work, the best electrochemical response for PAR was obtained using HAc/Ac^−^ buffer (Appendix A). Notably, in ammonia buffer (pH 8.5), a significant shift toward more negative potentials and a decrease in current signal were observed. This behavior is attributed to the easier deprotonation of PAR in basic media, resulting in its deprotonated form, which is less adsorptive and potentially less electrochemically active [65]. On the other hand, when using H_2_SO_4_ as the supporting electrolyte, a significant shift toward more positive potentials was observed, likely due to the extremely acidic environment promoting PAR degradation. The signal may correspond to p-aminophenol, a known acid hydrolysis product of paracetamol, which is electroactive and can be detected by voltammetric techniques within this potential range [66,67].

The effect of pH on the current response of 2 µM PAR was evaluated using the GCE/ph/ERGO sensing interface in 0.10 M HAc/Ac^−^ buffer over the range 3.50–5.50 (Appendix A). The maximum current was observed at pH 5. This behavior is likely due to a balance between proton availability and electron transfer efficiency, as the neutral form of PAR—which predominates below pKa of 9.46—is more electroactive at pH 5. At low pH values, excessive protonation of either PAR or the electrode surface may hinder electron transfer or promote side reactions. Conversely, at high pH values, the deprotonated form of PAR may be less electroactive or adsorptive on the electrode surface (see Appendix A for reaction). As the pH increases from 3 to 5, the oxidation signal improves, probably due to enhanced deprotonation of the phenolic hydroxyl group, which facilitates a proton-coupled electron transfer (PCET) mechanism [68]. Although paracetamol is expected to remain largely protonated and electrochemically active across the pH range tested, the observed current signal trend suggests a strong influence of the electrode surface properties. As supported by previous studies on molecule–surface interactions, changes in the surface charge can affect local interface behavior through the charge regulation phenomenon [69,70]. Specifically, the analyte and the surface dynamically influence each other’s protonation state, potentially shifting the apparent pKa of the analyte. This may explain the enhanced signal at pH 5 and its subsequent decline at pH 5.5, even though bulk protonation remains relatively constant. Further studies will be conducted to determine if there is a surface effect on the displacement of pKa.

### 2.7. Analytical Performance Evaluation

#### 2.7.1. Linear Range and Limit of Detection

To obtain the linear range and the limit of detection (LOD) and quantification (LOQ), a calibration curve was fabricated with the GCE/ph/ERGO electrode by SWV under optimized conditions of 1 mM pNA and 1 mM NaNO_2_ in the electrografting process, 5 μL of the GO/NaNO_2_/HCl agent, and 0.10 M acid/acetate buffer pH 5. Figure 4A,B shows the voltammograms and calibration curve for 0.10–0.80 µM PAR. The LOD and LOQ were calculated using the standard deviation of the intercept, as previously reported [44,55,71,72,73,74,75], and the values obtained were 18.20 and 60.60 nM, respectively. The sensitivity from the slope of calibration plots was 2.46 µA µM^−1^. The analytical performance of the developed GCE/ph/ERGO sensor for PAR determination compared to other electrochemical surfaces is represented in Table 1. Compared to previously reported sensors for paracetamol detection, our sensor’s excellent performance indicates its potential for applications in pharmaceutical quality control. From Table 1, it is easy to deduce that it is one of the best sensors in terms of LOD.

#### 2.7.2. Repeatability and Reproducibility Studies

The repeatability of the proposed GCE/ph/ERGO electrode was assessed through the stability of a 0.30 µM PAR current signal in a series of 14 measurements with one electrode under optimized conditions. Figure 4C shows a constant peak current with a relative standard deviation (RSD) of 2.22%, which indicates good precision. To evaluate the sensor reproducibility, four different electrodes were prepared and used to determine the current of 0.30 µM PAR under optimized conditions (Figure 4D), resulting in a 7.64% RDS value. In addition, the stability of the sensor was tested by measuring the response 24 h after its modification, and no significant signal loss was found, validating the consistency and robustness of the fabricated sensor.

#### 2.7.3. Excipient Analysis

The influence of various excipients commonly used in PAR pharmaceutical formulations was assessed to evaluate the systematic applicability of the proposed electrochemical sensor. The excipient analysis was conducted using the SWV method under optimized conditions for the fabricated electrode, incorporating a 10- and 100-fold excess of excipients. This study aimed to determine the impact of 1 µM and 10 µM of talc, starch, stearic acid, magnesium stearate, silicon dioxide, and microcrystalline cellulose PH 101 excipients on the peak current of a 0.10 µM PAR signal. Appendix A shows that only magnesium stearate and microcrystalline cellulose PH 101 cause interference in the 10-fold excess addition and that stearic acid, magnesium stearate, and microcrystalline cellulose PH 101 cause interference in the 100-fold excess addition.

### 2.8. Accuracy and Application in Real Samples

The accuracy of the proposed methodology was evaluated using a generic pharmaceutical formulation of PAR (paracetamol 500 mg). The pharmaceutical dosage solution was prepared by dissolving a known amount of tablet powder in 100 mL of ethanol. Then, the tablet solution was diluted with 0.10 M acid/acetate buffer pH 5 to prepare the desired concentration of the pharmaceutical sample. The prepared tablet solution analysis was performed using SWV measurements by adding known concentrations of PAR standard to the real pharmaceutical sample by the standard addition method (Figure 5). The results obtained are in accordance with the labeled content on the tablet, obtaining a recovery percentage of 105% (see Table 2), a slight deviation attributed to minor matrix-induced signal enhancement or analytical variability, which falls well within the AOAC’s accepted range for analytical accuracy. These results were compared with HPLC analysis made for the same tablet powder of PAR pharmaceutical dosage by the standard addition method, obtaining a 98% recovery percentage. Therefore, based on the results obtained, we can conclude that the fabricated sensor has a similar performance to the certified HPLC analysis and verifies the effective reliability and validity of the sensor for feasible screening analysis in the pharmaceutical industry.

## 3. Materials and Methods

### 3.1. Chemicals and Materials

All reagents are of analytical grade. Ultra-Highly Concentrated Single-Layer Graphene Oxide (C_x_O_y_H_z_, 6.2 gL^−1^) was purchased from Graphene Supermarket (Calverton, NY, USA). Acetaminophen (PAR, CH_3_CONHC_6_H_4_OH, 98%), 4-nitroaniline (pNA, O_2_NC_6_H_4_NH_2,_ 99%), sodium nitrite (NaNO_2_, 97%), potassium chloride (KCl, 99%), potassium ferrocyanide (K_4_[Fe(CN)_6_], 98.5%) and ferricyanide (K_3_[Fe(CN)_6_], 99%), disodium hydrogen phosphate (Na_2_HPO_4_, 99%), sodium dihydrogen phosphate dihydrate (NaH_2_PO_4_·2H_2_O, 99%), sodium hydroxide (NaOH, 97%), sodium acetate (CH_3_COONa, 99%), ammonium chloride (NH_4_Cl, 99.5%), boric acid (H_3_BO_3_, 99.5%), acetic acid (CH_3_CO_2_H, 99.99%), fuming hydrochloric acid (HCl 37%), phosphoric acid (H_3_PO_4_, 85%), nitric acid (HNO_3_, 65%), sulfuric acid (H_2_SO_4_, 95–97%), ammonia (NH_3_, 99.98%), acetonitrile (CH_3_CN, 99.9%), and absolute ethanol (EtOH, CH_3_CH_2_OH, 99.5%) were purchased from Sigma-Aldrich (Darmstadt, Germany). The acetic/acetate (HAc/Ac^−^) buffer solution used in this work contained 0.10 M CH_3_COONa and 0.10 M CH_3_CO_2_H, adjusted to pH 5 with NaOH, and the phosphate buffer solution (PBS) contained 0.10 M Na_2_HPO_4_, 0.10 M NaH_2_PO_4_, and 0.10 M H_3_PO_4_, adjusted to pH 7 with NaOH. The ferro/ferricyanide solution was prepared using 5 mM K_4_Fe(CN)₆, 5 mM K_3_Fe(CN)_6_, 0.50 M KCl, and PBS. Water miliQ (Burlington, MA, USA) (˃18.30 MΩ·cm) was used for the preparation of all solutions and material cleaning. All the measurements were conducted in an extra pure nitrogen (N_2_, 99.995%) atmosphere at ambient temperature (23–26 °C).

### 3.2. Instrumentation

All electrochemical measurements were performed with a 621E model potentiostat from CH Instruments Inc. (Austin, TX, USA) in an electrochemical cell using a three-electrode system: a glassy carbon working electrode (GCE, CHI104, 3 mm diameter) or GCE/ph/ERGO, saturated calomel reference electrode (SCE), and auxiliary platinum wire electrode. Electrochemical impedance spectroscopy (EIS) measurements were performed in a frequency range from 0.1 to 100,000 Hz with an amplitude of 5 mV. The prepared electrodes were characterized by scanning electron microscopy (SEM) using FE-SEM with a FEI Quanta FEG 250 model from ThermoFisher Scientific (Waltham, MA, USA).

### 3.3. Fabrication of Sensing Interfaces for Detection

Before use, GCE was polished with a slurry of 0.05 µm alumina powder and then rinsed abundantly with water and dried at room temperature to achieve a clean surface. The preparation of the p-aminophenyl-modified GCE (GCE/pNH_2_) used is reported in the literature [29]. A total of 1 mmol L^−1^ NaNO_2_ was added to a cooled solution of 1 mmol L^−1^ of *p*-nitroaniline (pNA) in 0.50 M aqueous HCl and left in an ice bath for 10 min to generate the nitrophenyl diazonium salt. Then, the electrografting process was conducted in this solution by cyclic voltammetry (CV) from 0.50 to −0.50 V at 100 mV s^−1^ for 2 cycles. The electrode was rinsed with water, acetonitrile, and water. The nitro groups in the modified surface were reduced to amino groups in 1 M KCl ethanolic solution (90%H_2_O/10%EtOH) using CV from −0.40 to −1.50 V at 50 mV s^−1^ for 3 cycles. The electrode was rinsed with water, and an aliquot of 5 µL of a solution of 0.50 M HCl, 5 mM NaNO_2_, and 0.62 g L^−1^ graphene oxide (GO) was deposited on the GCE/pNH_2_ electrode surface and allowed to dry at room temperature to convert the external amino group -NH_2_ to -N_2_ by diazotization, which was displaced by GO due to instability to obtain GCE/ph/GO. The reduction of GO to electrochemically reduced graphene oxide (ERGO) was performed by CV from 0.00 to −1.60 V at 100 mV s^−1^ for 7 cycles in 0.50 M NaCl solution to obtain GCE/ph/ERGO.

### 3.4. Electrochemistry Measurement of PAR

Square wave voltammetric measurements (SWVs) were performed with a scanning potential from 0.3 to 0.70 V using a step potential of 4 mV, pulse amplitudes of 25 mV, and a frequency of 15 Hz. All measurements were conducted with oxygen removal. Repeated blank measurements were recorded in 10 mL of HAc/Ac^−^ buffer until a stable background current was obtained. Individual calibrations of PA were performed in 0.10 M HAc/Ac^−^ buffer at pH 5 by adding increasing concentrations of the analyte in triplicate.

### 3.5. Sample Preparation

The real sample analysis was performed by SWV and high-performance liquid chromatography (HPLC). PAR commercial tablets were purchased from a local pharmaceutical store, each containing 500 mg of PAR. The tablets were weighed and finely powdered in a mortar and pestle. For HPLC analysis, to the corresponding weight, 25 mL of methanol LiChrosolv^®^ grade was added to obtain a 33.08 mM PAR sample solution, which was shaken and filtered through a 0.45 µm syringe membrane filter, and an appropriate volume of the PAR solution was diluted to 50 mL with a 10 mM PBS pH 7.5 buffer and methanol mixture. The measurement conditions were 1 mL min^−1^ flow rate, 10 µL injection volume, 8 min run time, 220 nm UV/vis detector, and 80 PBS/20 methanol mobile phase. For SWV measurements, a known amount of PAR tablet powder was dissolved in 100 mL of ethanol, and then the tablet solution was diluted with 0.10 M acid/acetate buffer pH 5 to obtain a 0.10 mM PAR sample solution. The prepared tablet solution analysis was performed using SWV measurements by adding known concentrations of the prepared PAR standard to the real pharmaceutical sample by the standard addition method.

## 4. Conclusions

The study describes the fabrication and characterization of an electrochemically reduced graphene oxide immobilized in a glassy carbon electrode sensor with electrografting of phenyl diazonium salts. The combination was chosen to improve the sensor surface stability, conductivity, and sensitivity toward phenolic compounds like paracetamol. The resulting robust and sensitive surface showed an obvious increase in the PAR detection signal compared to unmodified electrodes or only ERGO electrodes. Under optimized conditions, PAR was detected in the presence of interfering excipients in 10- and 100-fold excess, with good RSD results in repeatability and reproducibility tests. The linear range and LOD obtained are comparable with those found in the literature for PAR detection with voltammetric techniques. The new sensing platform proved to be sensitive in pharmaceutical matrices, with recovery close to 100%, comparable with accepted techniques such as HPLC. Although this work focused on PAR detection due to its pharmaceutical relevance and redox activity, the electrode modification could be extended to other electroactive drugs with similar structures, such as catecholamines or other phenolic pharmaceuticals, such as Ibuprofen, Caffeine, and Rutin, among others. Finally, while the sensor demonstrates excellent sensitivity toward paracetamol in pharmaceutical formulations, and its design is intended for analytical detection rather than pollutant removal or membrane-based purification, it could be potentially used to detect PAR as an emergent contaminant in aqueous samples, provided that an analysis of possible interferents present in wastewater is made.

## Data Availability

The raw data supporting the conclusions of this article will be made available by the authors upon request.

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
