# Peer review of "Electrochemically Reduced Graphene Oxide Covalently Bound Sensor for Paracetamol Voltammetric Determination"

_ijms, 2025, doi:10.3390/ijms26094267_

Round 1
Reviewer 1 Report
Comments and Suggestions for Authors
The manuscript titled “Electrochemically reduced graphene oxide covalently bind sensor for Paracetamol voltammetric determination” is interesting. The present work develops an electrochemical sensor based on a glassy carbon electrode (GCE) modified with phenyl diazonium salts (ph) and electrochemically reduced graphene oxide (ERGO) obtaining (GCE/ph/ERGO) for the detection of a widely used pain reliever drug, paracetamol (PAR), in pharmaceutical matrices using square wave voltammetry (SWV). The manuscript can be considered for publication after addressing few comments during the revision, as indicated below.
- Please re-write the abstract section to highlight the important findings of the article.
- Scheme 1 should be redrawn, as its image quality is not at standard.
- It is required to emphasize the novelty of the current work in comparison to previously published work in the same field.
- Please mention the purity and place of procurement of chemicals used.
- Is authors calculated electroactive surface area of WEs, before and after modification? The following literature can be used to perform the calculations. Microchemical Journal Volume 201, 2024, 110731; Ecotoxicology and Environmental Safety Volume 282, 2024, 116701. Inorganic Chemistry Communications, 172, 2025, 113764; Ecotoxicology and Environmental Safety Volume 253, 2023, 114694;
- What is the significance of electrolyte pH? Have you investigated the number of electrons/protons involved in the reaction mechanism?
- Have you examined the sensor's stability and reproducibility?
- Real sample analysis has to be explained in detail.
- How did you calculate the LOD of the sensor? Table 1 should be revised with more recent studies.
Author Response
We highly value the reviewer's recommendations, which have markedly enhanced the quality of the manuscript. Each comment has been thoroughly addressed, as outlined below.
Comments 1: Please rewrite the abstract section to highlight the important findings of the article.
Response 1: We have restructured the abstract to highlight the results: the sensitivity, reproducibility, and robustness of the modified electrode, and cleaned up the text for better understanding.
Comments 2: Scheme 1 should be redrawn, as its image quality is not at standard
Response 2: A new image assigned as Scheme 1 was re-drawn with molecular modeling software, and it is shown on Page 7, line 277.
Comments 3: It is required to emphasize the novelty of the current work in comparison to previously published work in the same field.
Response 3: We appreciate the reviewer request for clarification regarding the novelty of our work. In the revised manuscript, we explicitly highlighted the unique aspects of our approach. Specifically, in Sections 1. (Page 3, lines 102 – 105), and 2.6.1. (Page 9, lines 361 – 364), we emphasize that the combination of phenyl diazonium salts and ERGO on a glassy carbon electrode for paracetamol detection has not been previously reported. We present an updated Table 1 (Page 9, line 380) for a comparison of our sensor and recent studies. The corrected text highlights the increase in current obtained for the analyte compared to the unmodified electrode (see abstract) and the comparison with other systems (Table 1)
Comments 4: Please mention the purity and place of procurement of the chemicals used
Response 4: The purity and abbreviations of the used reagents have been added in the “Chemicals and Materials” section (Page 12, lines 423 – 434).
Comments 5: Have the authors calculated the electroactive surface area of WEs, before and after modification? The following literature can be used to perform the calculations. Microchemical Journal Volume 201, 2024, 110731; Ecotoxicology and Environmental Safety Volume 282, 2024, 116701. Inorganic Chemistry Communications, 172, 2025, 113764; Ecotoxicology and Environmental Safety Volume 253, 2023, 114694;
Response 5: We have added the electroactive surface area of modified surfaces in Section 2.3. (Page 6 and 7, lines 234 – 260).
Comments 6: What is the significance of electrolyte pH? Have you investigated the number of electrons/protons involved in the reaction mechanism?
Response 6: We have expanded the explanation related to the effect of the type of supporting electrolyte and its pH on the electrochemical oxidation of paracetamol in Section 2.5.3. Specifically, we have added on Page 8, lines 310 – 312 and 317 - 329; and on Page 9, the lines 332 – 349. Additionally, we have added Figure S5 in the supporting information for easy comprehension of the pH effect on the paracetamol molecule. Our manuscript shows a scheme that accounts for the reported oxidation of acetaminophen, which corresponds to an oxidation involving 2 electrons and two protons (Scheme 1). This reaction would be expected to be favored at any pH where the majority analyte species is deprotonated, but a clear maximum is observed at pH 5. This indicates that there is some relationship between the pH of the polarized surface and the affinity for the electrolyte. It is hoped in the future to perform calculations of the type described in reference 70 to determine if there is an effect of surface area on a shift of the pKa of paracetamol under the conditions described.
Comments 7: Have you examined the sensor's stability and reproducibility?
Response 7: We demonstrated the accuracy of the developed sensor through repeatability and reproducibility tests that prove the stability in the measurement of paracetamol with the developed sensor, obtaining an RSD of 7.64% for reproducibility tests and 2.22% in repeatability tests. The reviewer can find the details in Section 2.6.2. (Page 9, lines 366 – 374). Reproducibility tests were performed by modifying 4 different electrodes and their response to paracetamol was analyzed, obtaining an RSD of less than 8%. In addition, the stability of the sensor was tested by measuring the response 24 hrs after its modification, and no significant signal loss was found.
Comments 8: Real sample analysis has to be explained in detail
Response 8: We have explained the real sample analysis in Section 2.7. “Accuracy and application in real samples”. Additionally, we have polished the details of sample preparation for HPLC and SWV detection in the methodology in Section 4.5. (Page 13, lines 483 – 488).
Comments 9: How did you calculate the LOD of the sensor? Table 1 should be revised with more recent studies.
Response 9: We explain the mathematical calculation in Section 2.6.1. (Page 9, lines 356 – 358) in more detail. Additionally, we have updated the literature shown in Table 1 on Page 10 for more recent studies in paracetamol detection.

Reviewer 2 Report
Comments and Suggestions for Authors
The authors reported a sort sensor based on a glassy carbon electrode (GCE) modified with phenyl diazonium salts (ph) and then electrochemically reduced to graphene oxide (ERGO) obtaining (GCE/ph/ERGO).
The authors presented the results in the way to prove their expected outcome. However, there are unclear issues that need to be clarified before publishing. Please find below some comments/suggestions meant to help in improving the quality of the manuscript.
- The exact purpose of the preparation of such electrode was not clearly presented; in consequence , the exact importance of such electrode was not fully understood. Additionally, the authors reffered just to the detection of paracetamol. The question arising is that the prepared electrode is meant to be applicable just for paracetamol or there are also concerns for the detection of other drugs. The authors are adviced to provide details
- The authors stated that the developed electrode is sensitive to the presence of drugs. The exactly the source or media to be detected was not clearly understood. Can this electrode play a role like membrane for water purification nearby pharmaceutical plants? the authors are requested to provide details.
- The deposition of the material was done electrochemically. However, no detail on the deposited layer was provided scuh as thickness, etc. The authors provided SEM images but unfortunatelly, the images were not convinsing so that to support any outcome. Maybe SEM or other analysis concerning the deposited layer should be supported by literature or other similar studies.
- Within Figure 1 was illustrated the schematic procedure; however, the way to control the growing polymer over the surphace was not clearly undertood. The authors are requetsed to provide details.
- Table 2: The authors presented the Recovery percentage obtained from pharmaceutical real sample analysis. however a results of 105% recovery was listed and the reason for such results was not clearly understood. It looks like a statistical erros. The authors are requested to provide additional explanation.
Author Response
We appreciate the feedback provided by our reviewers, as it helps us improve our work. The following sections will address each comment in detail:
Comments 1: The exact purpose of the preparation of such an electrode was not clearly presented; in consequence , the exact importance of such electrode was not fully understood. Additionally, the authors referred just to the detection of paracetamol. The question arising is whether the prepared electrode is meant to be applicable just for paracetamol or there are also concerns for the detection of other drugs. The authors are advised to provide details.
Response 1: We have added a paragraph clarifying the rationale behind the electrode design, concretely in Sections 1 (Page 2 and 3, lines 90 – 99), and 5. (Page 13, lines 492 – 495). The combination of phenyl diazonium salts and reduced graphene oxide was chosen to synergistically improve surface stability, conductivity, and sensitivity toward phenolic compounds like paracetamol. Moreover, not only does ERGO improve electrode performance, but the way it is deposited improves the current response of the analyte (shown by the difference between our electrode and GCE and ERGO electrode). The modification method preserves conductivity and surface accessibility, which is demonstrated experimentally with CV and EIS experiments.
Although this work focused on paracetamol as a model analyte due to its pharmaceutical relevance and redox activity, the electrode modification could be extended to other electroactive drugs with similar structures. Recently, we evaluated this surface modification for another phenolic pharmaceutical drug with satisfactory results, which will be considered for future publications, but this indicates that the sensor is reliable for oxidizable molecules of similar structure but is not selective.
Comments 2: The authors stated that the developed electrode is sensitive to the presence of drugs. The exact source or media to be detected was not clearly understood. Can this electrode play a role like a membrane for water purification near pharmaceutical plants? The authors are requested to provide details.
Response 2: We would like to clarify that the developed electrode was designed specifically for the electrochemical detection of paracetamol in pharmaceutical formulations, and potentially in aqueous samples. The intended application is as a sensing platform rather than a water purification membrane. While the surface modification may have potential for adsorption or interaction with pharmaceutical pollutants, the electrode was not engineered or evaluated for use in remediation processes. Also, our electrode is not selective. We have added clarification in the revised manuscript to distinguish between analytical detection and environmental removal applications in Section 5. (Page 13, lines 501 – 504).
Comments 3: The deposition of the material was done electrochemically. However, no detail on the deposited layer was provided, such as thickness, etc. The authors provided SEM images, but unfortunately, the images were not convincing so that to support any outcome. Maybe SEM or other analyses concerning the deposited layer should be supported by literature or other similar studies.
Response 3: We have added the electroactive surface area of modified surfaces in Section 2.3. (Page 6 and 7, lines 234 – 260). More focused studies on the surface coverage could be performed in the future, focused on the atomic force microscopy (AFM) technique that would allow us to calculate the thickness of the modified diazonium layer on the GCE surface, as reported by Olguin et. al. in their study (10.3390/molecules26061631).
Comments 4: Within Figure 1 was illustrated the schematic procedure was illustrated; however, the way to control the growing polymer over the surface was not understood. The authors are required to provide details.
Response 4: We have added additional information in Sections 2.1. (Page 3, lines 120 – 128) and 2.3. (Page 5, lines 195 – 198) in the manuscript. We explain the carefully monitored parameters used for the grafting process. We state that the grafting strategy employed in this work was intentionally designed to minimize multilayer formation, a common challenge in diazonium electrografting. Using a low diazonium concentration (1 mM), limiting the potential to –0.5 V, and restricting the procedure to only two voltammetric cycles effectively reduced the formation of excess aryl radicals. The choice of 4-nitroaniline as the precursor further contributed to layer control due to the electron-withdrawing nature of the nitro group, which reduces the reactivity of the aryl radical toward further coupling. The good results obtained with CV and EIS demonstrate that mainly a single layer of “bridges” is obtained.
Comments 5: Table 2: The authors presented the Recovery percentage obtained from pharmaceutical real sample analysis. However, a result of 105% recovery was listed, and the reason for such results was not clearly understood. It looks like a statistical error. The authors are requested to provide additional explanation.
Response 5: We appreciate the reviewer’s attention to the recovery values. The 105% recovery falls within the AOAC's accepted range (95–105%) for analytical accuracy. Slight deviations above 100% are commonly observed in real sample analysis and may be attributed to minor matrix-induced signal enhancement or analytical variability within acceptable limits. The result suggests that the method slightly overestimated the analyte concentration, yet it remains within accepted analytical performance criteria. More importantly, this result supports the method's suitability for paracetamol determination in pharmaceutical formulations. However, we have added an explanation for this result in Section 2.7. (Page 11, lines 407 – 409).

Reviewer 3 Report
Comments and Suggestions for Authors
The manuscript presents a well-structured study demonstrating the successful fabrication of a sensitive and reproducible GCE/ph/ERGO sensor for paracetamol detection, supported by analytical validation. However, there are several areas that could be further improved to enhance the review before publication.
Comments:
- The equation numbering in the manuscript is inconsistent. Specifically, Equation (2) is used twice in different contexts.
- Chemical purity should be mentioned for all precursors.
- Related papers about Paracetamol, introducing the graphene and voltammetric techniques need to be cited with some recent supporting articles, which can be supported by reference works (DOI: 10.1039/D4EN00454J; doi.org/10.1016/j.colsurfa.2022.128740; doi.org/10.1016/j.compositesb.2022.109847; doi.org/10.1016/j.mtchem.2024.101896)
- Please explain how the number of electrons and protons involved in the electrochemical reaction.
- What are the electrodes used for elechemical studys?
- Materials characterization part should be improved for example XRD, UV-vis, FTIR and .
- Recommad to propose a possible electrochemical oxidation mechanism of paracetamol at the GCE/ph/ERGO.
- The current response for acetate buffer in Figure S3.A looks strange compared to what’s shown in Figure S3.B.
- Electrochemical active surface area (ECSA) of all electrodes should be evaluated
- In the EIS study, the X(real part) and Y (imaginary part) scale of the Nyquist plots are not equal.
- Some of the references are outdated. Please include more recent studies to show the latest advancements in this area.
Author Response
We are grateful for the reviewer's comments, which contribute to the enhancement of our work. Each comment has been addressed comprehensively, as outlined below:
Comments 1: The equation numbering in the manuscript is inconsistent. Specifically, Equation (2) is used twice in different contexts.
Response 1: We have reorganized the equations corresponding to redox reactions in a single paragraph for easy understanding, in Section 2.1. (Page 4, lines 159 – 162).
Comments 2: Chemical purity should be mentioned for all precursors
Response 2: The purity and abbreviations of the used reagents have been added in the “Chemicals and Materials” section (Page 12, lines 423 – 434).
Comments 3: Related papers about Paracetamol, introducing the graphene and voltammetric techniques, need to be cited with some recent supporting articles, which can be supported by reference works (DOI: 10.1039/D4EN00454J; doi.org/10.1016/j.colsurfa.2022.128740; doi.org/10.1016/j.compositesb.2022.109847; doi.org/10.1016/j.mtchem.2024.101896)
Response 3: We have updated Table 1 on Page 10 with more recent studies related to paracetamol voltammetric detection in pharmaceutical samples.
Comments 4: Please explain how the number of electrons and protons involved in the electrochemical reaction.
Response 4: We have added a brief explanation for every electrochemical reaction involved in the study in terms of electron-proton exchange. Specifically, we added equations detailing the electrons and protons involved in the electrografting process in Section 2.1. (Page 4, lines 159 – 162); we polished the explanation for the electrochemical reduction of p-nitrophenyl to p-aminophenyl in the surface layer in Section 2.1. (Page 4, lines 166 – 169); in Section 2.3. (Page 5, line 195) we added the electrons involved in the redox reaction of the probe, and we added an explanation for the electrochemical oxidation of paracetamol in Section 2.4 (Page 7, lines 264 – 269).
Comments 5: What are the electrodes used for electrochemical studies
Response 5: We have added a brief detail of the working electrode used for organic molecules detection in Section 1. (Page 2, lines 56 – 59). As for the electrodes used in this study, it is specified in Section 4.2. (Page 12, line 442 – 449).
Comments 6: Materials characterization part should be improved, for example, XRD, UV-vis, FTIR, and
Response 6: We appreciate your suggestion. Unfortunately, UV-vis and XRD techniques are not applicable techniques for characterization of this surface. The FT-IR technique could be used to characterize functional groups on the modified surfaces; however, we experienced drawbacks when performing it due to the thickness of the modified glassy carbon electrode. Another technique that could be performed is Raman for the characterization of modified GO and ERGO on the surface, however, this technique would not help us to interpret what is happening on the active surface of the modified electrode. We are considering expanding to SEM and AFM studies and including surface characterizations by FT-IR and Raman, in the future.
Comments 7: Recommend to propose a possible electrochemical oxidation mechanism of paracetamol at the GCE/ph/ERGO.
Response 7: We have added a brief detail for the published electrochemical oxidation of paracetamol in Section 2.4 (Page 7, lines 264 – 269). Additionally, we have Scheme 1 showing the electrochemical reaction of paracetamol on Page 7, line 277.
Comments 8: The current response for acetate buffer in Figure S3.A looks strange compared to what’s shown in Figure S3.B.
Response 8: In our investigations, we express our results in electrochemical signal height, i.e., peak height. In both cases, in both Figure S3 and S4, the reported signal for PAR in acetate buffer is 4.6 and 4.8 µA, respectively. What the reviewer may find odd in Figure S3 is the Y scale, because the voltammograms were scaled to present them all together in one image, however, this does not change the value of the peak height for PAR in acetate buffer. That said, we have changed the plot of voltammetric responses of paracetamol in different buffers for better understanding in the rebranded Figures S4.A and S4.B.
Comments 9: Electrochemical active surface area (ECSA) of all electrodes should be evaluated
Response 9: We have added the electroactive surface area of modified surfaces in Section 2.3. (Page 6 and 7, lines 234 – 260).
Comments 10: In the EIS study, the X(real part) and Y (imaginary part) scales of the Nyquist plots are not equal.
Response 10: The scale was adjusted in the Nyquist plot so that the X and Y axes have a range of 0 - 5000. The inset of the plot was also adjusted so that both axes have a range of 0 - 80. This is shown in Section 2.3. (Page 6, line 208).
Comments 11: Some of the references are outdated. Please include more recent studies to show the latest advancements in this area.
Response 11: We have updated Table 1 on Page 10 with more recent studies related to paracetamol voltammetric detection in pharmaceutical samples. Additionally, we have reviewed the manuscript and replaced references of outdated studies.

Round 2
Reviewer 1 Report
Comments and Suggestions for Authors
Accept
Reviewer 2 Report
Comments and Suggestions for Authors
The authors have answered to the addressed queries, and the manuscript was updated accordingly.
Reviewer 3 Report
Comments and Suggestions for Authors
The authors have addressed my comments, and the manuscript is now suitable for publication in its present form.